# Re-Defining the Villain in *A Song of Ice and Fire* from the Aspect of Totemism

**Emrah Öztürk**

Department of Film Making and Broadcasting, Faculty of Communication, Near East University, North Nicosia 99010, Cyprus; emrah.ozturk@neu.edu.tr

**Abstract:** The confrontation between good and evil is one of the essential aspects of the fantasy genre. In George R.R. Martin's epic fantasy novel series *A Song of Ice and Fire* (ASOIAF, 1996–2011), he approaches this conception from a critical point of view. Whilst Martin creates deep and challenging characters in his novels, he introduces the White Walkers to the audience as almost one-dimensional, classic antagonists. It can therefore be questioned whether this contradicts his approach towards the medieval ethos. In order to answer this question, I will approach the narration from the aspect of totemism, and will use totemic signs and values for my analysis. Firstly, I will establish a relationship between totemism and the Old Gods. Conceptions such as 'sacred totem animal', 'totem as an emblem', 'restriction of incestuous intercourse', and 'spirits' will be useful for comprehending the Old Gods. Furthermore, I will try to analyze the narration with the elements of totemism. Lastly, in the light of this examination, I will try to explain what theWhite Walkers represent within the narration.

**Keywords:** fantasy; religion; totemism; *A Song of Ice and Fire*; George R.R. Martin; white walkers

---

## 1. Introduction

The confrontation between good and evil is one of the essential aspects of the fantasy genre. As Fredric Jameson claims, this conflict has a link with the medievalist ethos. He considers that the works of Tolkien and his contemporaries, as well as the *Harry Potter* series, share this Christian nostalgia (Jameson 2005, pp. 58–60). The claim is potentially valid for a great number of fantasy novels. Nonetheless, despite the fact that George R.R. Martin's epic fantasy novel series *A Song of Ice and Fire* (ASOIAF, 1996–2011) constructs a pseudo-medieval world, it can still be questioned whether such a claim can be made in this case. Basically, the series narrates the stories of characters whose desire is to claim the Iron Throne for themselves on an imaginary continent called Westeros. Even though there are familiar medieval fantasy elements in the story, such as dragons, swords, and sorcery, as well as imprisoned maidens, noble knights, brave heroes, and wild monsters, Martin, in fact, subverts the tropes of the genre. As Lauren S. Mayer states, Martin "makes his way through the volumes gleefully smashing his own and other medievalist texts' attempts at world creation" (Mayer 2012, p. 63).

There are no purely good protagonists in the story. Martin highlights the relativity of good and evil by saying "the villain is the hero of the other side"[1]. Values and measures of good and bad can change based on different perspectives. He also states that "the battle between Good and Evil is waged within the individual human hearts"[2], and makes it clear that his saga will not follow the

---

definitive ethos of the medieval period. Antagonists such as Cersie or Jaime Lannister all exist with their inner conflicts. Even though they might be accepted as sinners from the perspective of the gods, the audience can still feel sympathy for them. However, this situation does not apply for all villains in the story. Whilst the realm suffers from schemes, conspiracies, and wars, in the far north of the continent in the area "Beyond the Wall", the emergence of icy creatures called White Walkers (also known as The Others) threatens the existence of humanity. These creatures, unlike the other characters in the series, are not complex from the moral aspect. This is the first unusual point about them. White Walkers are the only figures in Westeros who represent an absolute evil. Whilst Martin creates deep and challenging characters in his novels, he introduces the White Walkers to the audience as almost one-dimensional, classic antagonists. It can therefore be questioned whether this contradicts with his approach towards the medieval ethos.

1    For Collider's interview with George R.R. Martin on 17 April 2011, see https://collider.com/george-r-r-martin-interview-game-of-thrones/ (last visited on 29 June 2020).

2    For AssignmentX's interview with George R.R. Martin on June 2011, see http://www.assignmentx.com/2011/interview-game-of-thrones-creator-george-r-r-martin-on-the-future-of-the-franchise-part-2/ (last visited on 29 June 2020).

The second unusual point is that the White Walkers have a direct relationship with nature, particularly with winter. Jameson states that fantasy is wedded to nature, and he points out the important function of nature for this genre: "Nature seems to function primarily as the sign of an imaginary regression to the past and to older pre-rational forms of thought" (Jameson 2005, pp. 63–64). For instance, in Narnia or Middle Earth, nature is portrayed as an enchanted place that is full of wonders. However, it is still vulnerable to the invasions of the antagonists. The hero must always 'scour away' the dark lord's reign and protect the land from evil. In *A Song of Ice and Fire*, however, White Walkers are, in a sense, the 'winter' itself. They bring the season with them.

If Martin does not follow the medievalist fantasy ethos and its Christian nostalgia, then how must we define good and evil in his secondary universe? Which values does Martin highlight when forming a connection between evil and winter? Moreover, if the conception of evil is relative, then what is the definition of a 'villain' in *A Song of Ice and Fire*?

In order to answer these questions, I will approach the narration from the aspect of totemism, and will use totemic signs and values for my analysis. The reason for establishing a connection between totemism and the narration is because totemism shows similarities with the belief in the 'Old Gods', which is the only nature-oriented religion in Westeros. During my analysis, I will benefit from signs such as 'totem animal', 'totem as name', 'totem as heraldic figure', or the restriction of incestuous intercourse. Therefore, belief in the Old Gods is the religion of the protagonists of the story (the Stark children). Additionally, it is the only religion that acknowledges the White Walkers.

My approach towards Martin's novel series might be criticized for two distinctive reasons. Firstly, George R.R. Martin makes no mention of totemism in his novels. Additionally, the usage of heraldry or names within the narration points to the feudal lifestyle of medieval Europe (Mat 2018, p. 02). In addition, Martin admits that he was particularly interested in Britain when writing his novel series[3]. As Martin states, there is a significant resemblance between the War of The Five Kings incident in *A Song of Ice and Fire* and an actual historical event called 'The Wars of the Roses' (1455–1485). Accordingly, the enmity between House Stark and House Lannister corresponds to the conflict between House York and House Lancaster. The event called the 'Red Wedding' is a reference to the Clan Douglas' Black Dinner incident (1440) in Scottish history. Additionally, the map of Westeros is an inverted version of the map of Britain[4]. Consequently, it can be said that Martin's fictional world was inspired by Britain's history rather than totemism.

In this regard, Europe's heritage of totemism must be considered. G.L. Gomme acknowledges totemism when he argues about Britain's customs, such as wearing animal skins in traditional ceremonies:

---

[3]　For more information, see the conference of Mr. Martin at Brown University on 23 October 2014: https://www.youtube.com/watch?v=TB5AU_bCZJg (last visited 29 June 2020).

[4]　For more information, please visit brilliantmaps.com: https://brilliantmaps.com/westeros/ (last visited 29 June 2020).

"There does not seem any fitting explanation other than that they are survivals of old clan festivals, which took place when clans were totem-clans".

(Gomme 1889, pp. 350–51)

Alongside the festivals, Gomme points out many other customs and traditions of Britain that originated from totemism. As Mario Alinei highlights, alongside Britain, the whole of medieval Europe was affected by totemism;

"Everywhere in Europe, modern folklore offers innumerable traces of 'totemism' (...) Kinship names, magic names, and Christian names thus form a recurrent, structural relationship, which can be interpreted as a diachronic sequence for one and the same category of 'sacred animals,' the ultimate origin of which can only be found in totemism as a universal ideological stage".

(Alinei 1985, p. 332)

Nonetheless, due to the debatable history of totemism, my examination could be determined as 'outdated'. This determination could be the second criticism directed towards my approach.

Gregory Forth states that the theory of totemism was 'virtually abolished' decades ago when Claude Levi-Strauss declared it an illusion (Forth 2009, p. 263). We can add more names to the list of those who criticized the theory, such as Max Weber, MirceaEliade, and Robert Jones. Ultimately, one can admit that Victorian totemism is no longer functional for contemporary scientific literature. However, Forth also considers the claim of Roy Willis: "(Totemism), despite being declared dead, obstinately refuses to 'lie down'" (Forth 2009, p. 263).

When we examine the current discussions about totemism, we can see that this belief system has been re-defined from an ecological perspective by particular academicians. In his research about the Keiyo community in Kenya, Samuel KipkemeiKigen highlights the new reception of totemism. Despite the fact that Victorian totemism is now outdated, there are still totemic communities, such as the Keiyo or Nage, that have survived. Deborah Rose emphasizes totemism's 'connectedness', 'mutual interdependence', and the 'significance of the lives of non-human species' (Rose 1998, p. 14). The concepts such as 'interconnectedness with nature', 'kinship with nature', or 'genetic kin in the great community of life' highlight totemism's nature-oriented existence (Kigen 2018, pp. 170–71). In this context, I will utilize the 'connectedness with nature' in my examination, which is another sign of totemism. The villains of the story, who are related to nature, might be interpreted from this perspective.

My research universe will be limited to the books of the series, which are *A Game of Thrones* (AGOT, Martin 1996), *A Clash of Kings* (ACOK, Martin 1999), *A Storm of Swords* (ASOS, Martin 2000), *A Feast for Crows* (AFFC, Martin 2005), and *A Dance with Dragons* (ADWD, Martin 2011). Martin's non-fictional books *The World of Ice and Fire* (TWOIAF, Martin et al. 2014) and *Fire and Blood* (FAB, Martin 2018) will also be accepted as resource materials. The adaptation of this series for television by HBO is out of the scope of this research.

## 2. White Walkers: Creatures of Winter

The seasons in *A Song of Ice and Fire* are different fromours. Summers can last for years and winters can be as long as a lifetime. The North of Westeros has a cold climate, which creates the appearance of permanent winter. Even in the summer, there could be snow. However, despite the cold, snow, and freezing weather, winter has yet to come to Westeros. For the northerners, winter represents more than just a season. Historical incidents such as The Long Night (TWOIAF, p. 31) give an indication about how dark the winter could be. One of the prominent characters, Jon Snow, identifies winter with the gods: "If they (gods) were real, he thought, they were as cruel and implacable as winter" (AGOT, p. 128). This comparison is not in vain. The winter is interpreted as a doomsday: "Fear is for the winter, my little lord, when the snows fall a hundred feet deep and the ice wind comes howling out of

the north ( . . . ) while direwolves grow gaunt and hungry, and the white walkers move through the woods" (AGOT, p. 167). House Stark's family words "Winter is coming!" are a warning, a reminder of this doomsday. Marcin Sankowski claims that this faith could be related with the 'wyrd' concept: "'Winter is coming' is fatalistic on the whole, reminding one of the Anglo-Saxon concept of wyrd, yet they seem to find some sort of consolation in the old faith" (Sankowski 2015, p. 83).

However, alongside this religious depth, winter is also functional in terms of dramatic purposes. For instance, in C.S. Lewis's *The Chronicles of Narnia: The Lion, The Witch and The Wardrobe* (1950), Jadis, the White Witch, rules the land called Narnia, and she casts a spell that inflicts eternal winter upon the realm. When the Pevensie children defeat Jadis, winter turns into spring. Similarly to Narnia, in *A Song of Ice and Fire*, winter is used as a representation of the 'time of chaos and darkness' as well. Spring is the season of childhood and rebirth, whilst winter is always the season of the dark lord. Correspondingly, the dark lords in this context are the White Walkers. When people mention winter, they mostly refer to the White Walkers. Fear of these icy creatures is not unfounded, as they are magical beings with the ability to resurrect the dead. Thus far in the series, they have appeared three or four times. However, Martin introduces them as the main danger, the real villain, and the real threat of the story.

In *Narnia*, Jadis was a hybrid witch (half jinn and half giant), and the winter lasted for 900 years because of her magic. In *The Lord of the Rings*, Saruman the White was a wizard who caused significant harm to the Fangorn Forest and the Shire. In *Earthsea*, Cob uses magic in order to manipulate the principles of nature for his own benefit. These magicians and their acts can be summarized with Jameson's sentences: "In modern fantasy, whose medieval Imaginary seems to be primarily organized around the omnipresence of magic, itself enlisted in the pursuit of power by the great magicians in their reenactment of that cosmic struggle between Good and Evil which, as we have seen, expresses the aristocratic ideologies of the medieval aesthetic" (Jameson 2005, p. 63).

In *A Song of Ice and Fire*, are the White Walkers magicians? Do they hunger for power like Saruman or Jadis? Are they seeking vengeance? The answer to these questions is simple: We do not know. Thus far, the author has not revealed anything about these creatures. The mystery of the White Walkers is one of the differences in the narration from other fantasy villains. Their intensions, their desires, and their backstories are completely unknown.

However, there is one thing that is certain about them: The White Walkers embrace winter. They have an organic and strong connection with this season. The concepts or words that Martin uses in order to describe them are similar to a description of a cold winter: 'Freezing weather', 'howling winds', 'blizzards', 'snow', 'ice', 'cold', 'mist', 'blue and white', 'milk and crystal', 'icicle and ice cracking', etc.[5] When they leave a place, the sky becomes clear and snow melts away, just like winter turns into spring.

Northerners, those who worship the Old Gods and believe that the winter is the doomsday, acknowledge the White Walkers on a religious level (like Satan or the Devil). Consequently, being connected with winter locates the White Walkers in this nature-oriented religion. In this vein, before I begin to examine these villains from the totemic aspect, I will try to interpret the belief in the Old Gods with totemic signs.

## 3. Belief in the Old Gods: Manifestation of Totemism?

The very first inhabitants of Westeros were a magical folk called the 'Children of the Forest' (TWOIAF, p. 27). Due to their nature-friendly character and child-like appearance, they were given this name by the First Men, the first human beings on the continent. The Children of the Forest embraced the countless and nameless gods of nature. They recognized the 'weirwood' trees (a tree species with white branches and red leaves) as sacred residences of gods. Faces were carved onto these

---

[5] A Wiki of Ice and Fire, 'Others'.https://awoiaf.westeros.org/index.php/Others#Appearance_and_Characteristics (last visited 30 June 2020).

trees in order to allow the gods to watch over them. Every pact, oath, funeral, wedding, and all other celebrations took place in front of these trees and, of course, in front of the gods as well. The First Men also embraced this belief and sustained it for thousands of years.

As can be seen, 'weirwood trees' are the center of this belief system. The concept of worshiping trees can be traced back to primitive and archaic religions. Sir James George Frazer stated that it is one of the oldest types of worship in history (Frazer 1912, p. 105). He provided many examples of such practices, such as those by the Celts, Germans, Slavs, and Ancient Greeks. For instance, at the time of the Roman Empire, the fig tree of Romulus was considered sacred (Ibid. p. 106).

The underlying concept behind the perception of trees being sacred originated in communities who believed that trees have souls and that they are actually living beings. Furthermore, some communities believed that the souls of the dead inhabit trees. In Middle Australia, members of the Dieri Clan think that some trees are actually transformed versions of their ancestors (Ibid. p. 107). Additionally, according to some clans, such as the islanders of Sioo in East India, trees are merely a stopover for souls. A soul has the freedom to leave the tree and return whenever it desires (Ibid. p. 110). Even today, as Gregory Forth states, the Nage people of the Indonesian island of Flores acknowledge particular trees as sacred and taboo (Forth 2009, p. 264). In addition, the work of Pandey and Pandey shows that trees are still believed to be residences of gods: "Peepal (Ficusreligiosa L.) is the most sacred tree in India. It is believed to bethe residence place of the triad—Brahma, Vishnu, and Mahesh (Shiva)" (Pandey and Pandey 2016, p. 139).

When the narration is examined in this context, it can be said that weirwood trees are similar to the clans' sacred trees. They are not just a different type of tree species, but they are ancient and special. They are portrayed as residences of gods in the narration as well. As one of the key characters, Bran Stark claims that they are alive and see dreams: "They dream tree dreams" (ACOK, p.53). Through the carved faces of trees, gods watch the humans and the world. "There was something wild about a godswood (...) you could feel the old gods watching with a thousand unseen eyes" (ACOK, p. 181).

The weirwood trees can also be interpreted with the 'spirit' concept in totemism. According to Emile Durkheim, the most important function of the spirit in totemism is to protect and look after the clan (Durkheim 1965, p. 300). This function can be apprehended by the connection of the spirits with birth phenomena within clans. Every new baby in the clan comes to the world with the strength of the spirit. It does not take shape in the body of an individual, but it resides in a particular object, such as a rock, a stone, or a tree. In this vein, weirwood trees can be seen as the residences of spiritual beings.

Along with the spirit concept, other signs of totemism might be applied to the narration. In totemism, every clan of the tribe chooses a totem that has ties with a mythical ancestor. Generally, the clan accepts an animal as its totem, but the totem could also sometimes be a plant, a tree, or another natural object (Ibid, pp. 103–5). In time, the totem becomes the symbol and identity of the clan. In *A Song of Ice and Fire*, every house has an emblem that serves a heraldic purpose; pictures of animals (bear, moose, horse, etc.) and pictures of plants or natural objects (trees, flowers, sea, lake, sky, etc.) are on the flags and shields. "Over their heads flapped the banner of the Starks of Winterfell: A gray direwolf racing across an ice-white field" (AGOT, p. 16). They can be seen as totemic figures as well. In this vein, the direwolf can be accepted as House Stark's totem animal. The emblem of the direwolf allows the Starks to dissociate themselves from others, but also gives them a persona. This is because the totem becomes the clan's symbol and identity. In various chapters, the Starks are mentioned as 'wolves' or 'direwolves'.

Even though the vast majority of people in Westeros believe in other religions, they still sustain several ancient customs; the use of a totem as a form of heraldry is one of these. For instance, House Baratheon's emblem is a stag, House Lannister's is a lion, House Tully's is a trout, House Arryn's is a falcon, and so on. Correspondingly, all houses identify themselves with their totemic animal/object: "You do not steal from the dragon, oh no. The dragon remembers" (GOT, p. 27), "when the stag and direwolf had joined . . . " (AGOT, p. 33), or "He had opened the city to the lions at the gate . . . " (GOT, p. 86).

In his study, G.L. Gomme gives examples from the totem-names of kings or lords in Britain: 'The Great Wild Cat', 'Sons of the Worm', 'Crawe the Crow', and so on (Gomme 1889, pp. 354–60). It is possible to see the same totem-name tradition in Australian tribes as well (Gibson and Gardner 2019, pp. 55–56). Parallel to this, characters in the narration also have aliases associated with their clan/house totems. Eddard Stark is also known as 'The Quiet Wolf', his sister Lyanna is called 'The She-Wolf', and his son Rob is named as 'The Young Wolf'. These totem-names can be seen with regard to other characters: Tywin Lannister is 'The Old Lion', Jaime Lannister is 'The Young Lion', Brynden Tully is the 'Blackfish', Jeor Mormont is 'The Old Bear', Maege Mormont is 'The She-Bear', and so on.

The conception of 'connectedness with nature' comes forth as one of the main characteristics of totemism. As Kigen quotes from James William Gibson, in totemism, "nature is an extension of human culture; even though humans and animals may no longer speak the same language, relationships between them are social and culturally imbued with spiritual significance" (Kigen 2018, p. 176). Humans and nature are two essential parts of a connection. In the narration, the descendants of the First Men who believe in the Old Gods have a rare gift called warging. Simply put, it is the temporal transformation of one's consciousness and soul into another's body. Those who have this ability are called 'wargs' or skin-changers. Wildling clans and some of the Stark children are gifted with this ability, especially Bran Stark. Warging is a connection with nature and with non-humans. A warg can see from the eyes of nature, feel the life inside the animals and plants, and can even taste the blood of a prey through the mouth of a beast; "the greatest of them (wargs) could wear the skins of any beast that flies or swims or crawls, and could look through the eyes of the weirwood trees as well" (ASOS, p. 105). When Bran Stark discovers his potential to use this ability, in order to learn how to warg, he journeys to a tree-wizard, the Three-Eyed Raven, who can see the world with 'a thousand eyes and one'.

As can be seen in the last two paragraphs, the similarities with totemic structure are not only limited tothe northerners' religion, but also reach to the entire continent tocertain degrees. In this context, I will examine the narration based on the moral system of totemism in the following chapter.

## 4. Transgressions

From the perspective of the moral system, there are two major prohibitions in totemism. The first one is killing or eating the sacred totem animal, and the second one is incestuous intercourse. Members of the clan must show respect to the totem. In addition, the only way of marriage is exogamy.

After the prologue chapter, the series starts with adead totem animal, a direwolf. Lord Eddard, the lord of House Stark, and his sons (Rob and Bran Stark, and illegitimate son Jon Snow) find their totem's dead body in the wilderness. Afterwards, they notice that the direwolf had given birth to six pups before she died. Lord Eddard decides to make the children responsible for these pups. From this moment, an emotional connection establishes between the Stark children and the direwolf pups. During the series, some of the direwolves die. The first one is unwillingly executed by Eddard Stark himself. The second direwolf dies at the incident of the Red Wedding. The enemies of Rob Stark, 'The Young Wolf', replace his head with his totem animal's head in order to humiliate him. Another direwolf is abandoned by her owner (Arya Stark). The deaths of totemic animals can be perceived as bad omens. They lead to series of unfortunate events for particular people.

The concept of incest in *A Song of Ice and Fire* is a vital subject. All of Westeros avoids engaging in such intercourse because it is forbidden in all religions of the realm: "Incest was a monstrous sin to both old gods and new, and children of such wickedness were named abominations in sept and godswood alike" (ACOK, p. 312). The free folk in the "Beyond the Wall", who believe in the Old Gods, are strict about this restriction. Ygritte, a member of the free folk, explains this subject to Jon Snow with these words: "A true man steals a woman from afar, t' strengthen the clan. Women who bed brothers or fathers or clan kin offend the gods, and are cursed with weak and sickly children. Even monsters" (ACOK, p. 284).

However, House Targaryen, the former rulers of the realm, were infamous for their incestuous marriages. Aegon the Conqueror conquered Westeros with his two sister-wives (FAB, Martin 2018,

p. 8). The Targaryens believed that marrying with sisters and brothers would sustain their pure blood; "The line must be kept pure" (AGOT, p. 28).

The dominion of the Targaryens ended with Robert Baratheon's rebellion, which occurred sixteen years before the narration begins. When Robert becomes king, he marries with Cersei from House Lannister. However, Cersei maintains an incestuous relationship with her twin brother Jaime. Consequently, all three children of King Robert in fact belong to the Lannister siblings. This prohibited relationship initiates and advances the events in the narration: Jaime throws Bran Stark out of the window of a tower at Winterfell because the boy witnessed his and his sister's act of incestuous intercourse. After several events, Eddard Stark learns the secret of Lannister siblings, and intends to reveal it. However, he is captured by King Joffrey 'Baratheon', and is ultimately executed. In response, to avenge his father's unjust execution, Eddard Stark's oldest son Robb calls on his banner men and rebels against the throne. All the northerners declare Rob as the King in the North. On the other hand, Stannis Baratheon, Robert's brother, also learns the truth and claims that he is the rightful king of Westeros. He sends letters across the realm, and tries to inform the lords that Joffrey is anillegitimate king born of incest. In this regard, Stannis gathers his army and attacks the capital city. At the same time, Robert's other brother, Renly Baratheon, also claims that he is the rightful ruler. In summary, because of Cersei and Jaime's secret incestuous relationship as well as their efforts to hide this secret, the realm descends into chaos. The War of the Five Kings occurs and Westeros begins to suffer.

Another example of incestuous intercourse occurs in the far north of Westeros, "Beyond the Wall". When Jeor Mormont, the Lord Commander of the Night's Watch, witnesses a 'wight' (a dead person who was reanimated by the White Walkers) who attempts to kill him at the Wall, he decides to travel to the far north and confront the White Walkers. While doing so, he visits another man, Craster, who lives alone with his wives and daughters in the Haunted Forest, which is a part of the Land of Always Winter. Craster marries his daughters and sacrifices his sons to the White Walkers. Thus, he is able to live in peace in his keep.

Craster's incestuous intercourse can be viewed from the perspective of the taboo conception. As Sigmund Freud points out, taboo is a combination of 'holy dread' (Freud 1919, p. 13). The conception of taboo includes the meaning of "the sanctity (or unclean lines) which results from a violation of prohibition." Craster violates the taboo of incest, and ultimately becomes unclean and a part of the taboo as well. The location of Craster's keep confirms this claim; he lives in the Haunted Forest, which is a place of 'otherness' and 'a space of danger and of strangeness' (Marques 2016, p. 39). Hence, living here as aninhabitant of the forest makes Craster dangerous and strange, just like the White Walkers/The Others, who he accepts as gods. He gives his boys to them and asks for protection. In return, he lives securely in his keep with his family; "I'm a godly man, and the gods keep me safe" (ACOK, p. 228), he says. In other words, Craster offers sacrifices to his gods in order to live in peace. As Brian Morris quotes from W. Robertson Smith, the underlying reason for making a sacrifice is to experience a mutual sharing between worshippers and their god (Morris 2004, p. 183).

White Walkers are hardly gods, but they still take Craster's boys. According to the wivesof Craster, these infant males turn into them; "Craster's sons. The white cold's rising out there, crow. I can feel it in my bones. These poor old bones don't lie. They'll be here soon, the sons" (ACOK, p. 355). Obviously, White Walkers do not kill Craster's sons; rather, they look after them, and accept them into their community.

As can be seen, two prohibitions of totemism are violated by characters in the narration. These violations present an alternative way of interpreting the struggles and journeys of the characters. According to the totemic values, the sinner is the one who disrespects the totem animal and/or engages in incestuous intercourse. Whilst some of the antagonists, such as Cersei, Jaime, Joffrey, Euron, and Craster, engage in acts of incest, others, such as Tywin Lannister, Roose Bolton, and Walder Frey, are responsible for the killing of sacred totem animals.

Now that it has been determined that the belief in the Old Gods is a manifestation of totemism as well as that totemic values find a place in the narration, we can view the White Walkers from the perspective of totemism.

## 5. Disconnectedness with Nature

Another name used for the White Walkers is 'The Others'. The phrase "Others take you!" is very common in the narration, and it is used as a curse many times. Indeed, the White Walkers are The Others of the continent, the ones kept out of the realm. Their return is prevented by a giant ice wall. They are the reason for the existence of the Night's Watch. They are the archenemy of humanity. White Walkers can be defined simply with the conception of otherness. As Marques states, "the notion that the other is someone to be feared, an opponent, guilty of all things, usually connected to the devil" (Marques 2016, p. 36). Others are always the source of evil and darkness. They are the unknown, distant, strange, outsiders, sinister, and unusual.

There is no need for the White Walkers to seek immortality, more power, or any other desires. Their moral choices do not make them villains. Perhaps this is why they do not have any moral conflicts. As time goes by, their numbers multiply in The Land of Always Winter with the dead, abandoned, those born of incest—such as Craster's sons—sacrificed peasant girls and boys, etc. They all join the army of The Others, which continually expands. When the winter really comes to Westeros, everybody will join them eventually. In other words, if they are able to cross 'The Wall', everybody will become 'The Others'. Perhaps the source of the fear that they give to the hearts of people is this oppositeness.

As it is known, the whole series is written from the points of view (POVs) of the various characters in different chapters. With these multiple POVs, the audience can connect with Martin's fictional world and see from the eyes of those characters. As I quoted before, Martin says that "the villain is the hero of the other side". However, the reader never has the opportunity to appreciate the POV of the White Walkers. Perhaps this is the key point for the answer; the White Walkers have no point of view.

From the aspect of totemism, the White Walkers represent disconnection. In the totemic cosmos, nature is a big family in which everything is connected. This is why warging is important in the narration. As previously mentioned, warging is a parallel to the idea of connectedness with nature. A warg's mind connects with other animals, beasts, and trees. They can look at the world from their perspective. It is similar to reading a chapter from the point of view of a character. Wargs can establish empathy with those non-human beings. However, this connection would be lost if the White Walkers brought the winter, snow, and coldness to the realm. With the winter apocalypse, due to freezing weather or frozen water, all life would be turned into ice, and people would lose their connections with nature. Without seeing, connecting, and integrating with each other, they would become alien.

## 6. Conclusions

In this research, I have tried to examine the conception of the villain in George R.R. Martin's epic novel series *A Song of Ice and Fire*. Whilst Martin constructs a pseudo-medieval world, he does not follow the 'medievalist fantasy ethos'. In particular, the conflicts of 'Good vs. Evil' occur differently from inother fantasy novels. The main villains of the story, White Walkers, have a strong link with nature, and it is believed that they will bring an apocalyptic winter upon Westeros.

In order to examine these antagonists, I have utilized totemism, which is a nature-oriented religious system. As a first step, I have compared a fictional religion from the narration with totemism. The belief in the Old Gods incorporates the gods of nature, and weirwood trees are acknowledged as sacred. Worshippers of this belief show respect to nature and accept animals or other natural objects as their emblems. They acknowledge the winter as a doomsday and they believe that one day, doom will come. House Stark's family words refer to this expectation: "Winter is coming". When all these cultural and religious elements are considered, it is plausible to suggest that the belief in the Old Gods is a manifestation of totemism. There are distinctive significations to help us to establish such a link. Concepts such as 'sacred totem animal', 'totem-names', or 'totem as heraldry' conform to this

interpretation. Totemism and totemic belief systems highlight the connectedness between humans and nature; therefore, the ability of warging can be seen as the representation of this totemic value within the narration.

Alongside the Old Gods, the moral values of totemism also relate with the narration. The first violation (killing of a sacred totem animal) is the inception point of the events; the deaths of direwolves are the previews of unholy and bad things to come. The second violation, incestuous intercourse, is the main dynamic that pushes the story forward, where the Targaryens' incestuousmarriages, Cersei and Jaime's secret relationship, and Craster's deviant family are distinctive examples of this situation.

In conclusion, when we look to the White Walkers with all of these outcomes, we can see that they represent 'disconnectedness with nature'. On the contrary, forthe protagonist, Bran Stark, who trains to warg into trees and other beasts in order to connect with the world and see from nature's eyes, White Walkers intend to bring coldness to Westeros. Pure good or pure evil characters belong to classic medieval fantasies' ethos. Nevertheless, in *A Song of Ice and Fire*, the real danger for human life is the disconnection from nature, and this is what the White Walkers represent.

**Funding:** This research received no external funding.

**Conflicts of Interest:** The author declares no conflict of interest.

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
