# Peer review of "Re-Defining the Villain in A Song of Ice and Fire from the Aspect of Totemism"

_religions, doi:10.3390/rel11070360_

Round 1
Reviewer 1 Report
the author has had the courage to rewrite his article, as it was rejected on a first vote. In addition, it has done very well, so that the text is now critically well supported and delves into its meaning.
Reviewer 2 Report
Revision looks sound.
This manuscript is a resubmission of an earlier submission. The following is a list of the peer review reports and author responses from that submission.
Round 1
Reviewer 1 Report
-Are there no more academic references about totemism, mainly current and particularly related to literature?
-What is the god discipline to which this article belongs ?, or what disciplines do you want to interrelate? It contributes nothing to any discipline or to the interrelation between them- Basing the analysis of Martin's works on an ancient academic literature does not seem the best way to approach the analysis
-Also Mircea Eliade criticizes the traditional view on totemism, particularly on Frazer and Freud (hence the need to incorporate more current academic literature on the subject)
-The author analyzes the narrative argument, but not its literary -and religious- images, of greater complexity and also richness
-The author concludes that the analysis of totemism and ancient religions can be helpful in contemporary environmentalism, but this is not the subject of the article and, therefore, does not demonstrate it empirically
-In the bibliography, a greater critical apparatus of the author and of the works analyzed is missed, as well as their religious roots
Reviewer 2 Report
This article seems to be doing one thing--a reading of a literary text--only to shift focus at the very end in the conclusion and switch to making claims about improving ecological economy and climate change. This was a bit jarring on reading the text. However, that said, this reviewer was completing the essay and wondering, as the work progressed, what the point of the observations was. So, something did emerge in the conclusion, but it was rather unexpected and not adequately supported. Should the author desire to revise and restructure the essay to make it more effective, I would recommend starting with a more directed introduction that makes many of the conclusion's claims so that the reader understands the reason for the analysis of totemism from the beginning. However, the earlier suggestion that understanding the place and function of totemism helps to interpret the logic behind the narration would, in itself, be a sufficiently interesting topic for a paper, particularly on this series, though there would, again, need to be some more salient point that was being argued and as the conceptual occasion for the rest of the more or less summary arguments in the essay when the literary work is discussed. If this would be a desirable focus, then this reviewer suggests a revised focus with a point--something like a discussion of the role of transgression in the narrative thrust of the novels or something like that to otherwise organize and direct the rest of the commentary that is presented in the essay in its current form. Beyond that, the discussion of totemism through the social scientists is very good and well-supported. The presentation of Freud's work on totemism is not as strong, and would suggest, at the very least, discussing the relation between the primal father and the symbolic father in the text since these two significantly and purposely diverse figures in Freud's account seem to be collapsed in the author's essay. The author also seems throughout the text to be entirely invested in a historical understanding of totemism, and it is important to remember that Freud's contribution to this discourse is through psychoanalysis and, thus, his arguments and discussions move in the direction of an understanding of totemism (and the prohibitions it reinforces) as a structural element of the social conscious and not so much as an actual historical event or construct. Last point: while the author does a good job discussing some of the elemental totemic moments in the literary work, such as the significance and endurance of the weirwood tree, and the role of the totem animal in family identity and endogamic structure (e.g. the direwolf as the totem animal for the Stark family), the connection of totemism to the practice of "warging" is much less adequately developed in the essay. Overall, an interesting read, but for a rather narrow audience, and issues related to the consistency and presentation of the central or overriding argument are the main issues with the essay.